# The Nucleolus and Its Interactions with Viral Proteins Required for Successful Infection

**DOI:** 10.3390/cells13181591

**Published:** 2024-09-21

**Authors:** José Manuel Ulloa-Aguilar, Luis Herrera Moro Huitron, Rocío Yazmin Benítez-Zeferino, Jorge Francisco Cerna-Cortes, Julio García-Cordero, Guadalupe León-Reyes, Edgar Rodrigo Guzman-Bautista, Carlos Noe Farfan-Morales, José Manuel Reyes-Ruiz, Roxana U. Miranda-Labra, Luis Adrián De Jesús-González, Moises León-Juárez

**Affiliations:** 1Laboratorio de Virología Perinatal y Diseño Molecular de Antígenos y Biomarcadores, Departamento de Inmunobioquímica, Instituto Nacional de Perinatología, Mexico City 11000, Mexico; josemanuel7111@gmail.com (J.M.U.-A.); luis_5m5@hotmail.com (L.H.M.H.); bezyqcb@hotmail.com (R.Y.B.-Z.); erguzman87@hotmail.com (E.R.G.-B.); 2Posgrado en Biología Experimental, Departamento de Ciencias Biológicas y de la Salud (DCBS), Universidad Autónoma Metropolitana-Iztapalapa, Mexico City 09310, Mexico; 3Laboratorio de Microbiología Molecular, Departamento de Microbiología, Escuela Nacional de Ciencias Biologícas, Instituto Politécnico Nacional, Mexico City 11340, Mexico; jorgecerna1008@gmail.com; 4Departamento de Biomedicina Molecular, Centro de Investigación y de Estudios Avanzados del Instituto Politécnico Nacional (CINVESTAV-IPN), Mexico City 07360, Mexico; leebeydengue@yahoo.com.mx; 5Laboratorio de Nutrigenética y Nutrigenómica, Instituto Nacional de Medicina Genómica (INMEGEN), Mexico City 14610, Mexico; greyes@inmegen.gob.mx; 6Departamento de Ciencias Naturales, Universidad Autonoma Metropolitana (UAM), Unidad Cuajimalpa, Mexico City 05348, Mexico; carlos.farfan@cinvestav.mx; 7Centro Médico Nacional “Adolfo Ruiz Cortines”, Instituto Mexicano del Seguro Social (IMSS), Veracruz 91897, Mexico; jose.reyesr@imss.gob.mx; 8Departamento de Ciencias de la Salud, Universidad Autónoma Metropolitana-Iztapalapa, Mexico City 09310, Mexico; roxml@xanum.uam.mx; 9Unidad de Investigación Biomédica de Zacatecas, Instituto Mexicano del Seguro Social, Zacatecas 98000, Mexico

**Keywords:** nuclear bodies, ribosome biogenesis, Cajal bodies, nucleolus, promyelocytic leukemia nuclear bodies, transcription, non-coding RNAs

## Abstract

Nuclear bodies are structures in eukaryotic cells that lack a plasma membrane and are considered protein condensates, DNA, or RNA molecules. Known nuclear bodies include the nucleolus, Cajal bodies, and promyelocytic leukemia nuclear bodies. These bodies are involved in the concentration, exclusion, sequestration, assembly, modification, and recycling of specific components involved in the regulation of ribosome biogenesis, RNA transcription, and RNA processing. Additionally, nuclear bodies have been shown to participate in cellular processes such as the regulation of transcription of the cell cycle, mitosis, apoptosis, and the cellular stress response. The dynamics and functions of these bodies depend on the state of the cell. It is now known that both DNA and RNA viruses can direct their proteins to nuclear bodies, causing alterations in their composition, dynamics, and functions. Although many of these mechanisms are still under investigation, it is well known that the interaction between viral and nuclear body proteins is necessary for the success of the viral infection cycle. In this review, we concisely describe the interaction between viral and nuclear body proteins. Furthermore, we focus on the role of the nucleolus in RNA virus infections. Finally, we discuss the possible implications of the interaction of viral proteins on cellular transcription and the formation/degradation of non-coding RNAs.

## 1. Introduction

Viruses are considered intracellular parasites, and over the years, they have developed mechanisms and strategies to regulate processes such as the cell cycle, apoptosis, autophagy, immunological processes, and metabolism. Additionally, viruses can change the distribution of cellular organelles, such as the mitochondria, endoplasmic reticulum, and Golgi complex, to efficiently carry out their replicative cycle [1,2]. Some viruses invade the nucleus of the host cell to carry out their replication. Indeed, it is now well established that viruses can interact with both cellular protein elements and phospholipids. Furthermore, viruses can modify the structures of the nucleus and interact with nuclear pores, which control the nucleocytoplasmic trafficking of viral proteins. This is achieved through an interaction with nucleocytoplasmic transport proteins, such as importins and exportins, to allow access to the nucleus and nucleolus. To enable transport between the nucleus, or nucleolus, and the cytoplasm, viral proteins must contain nuclear localization sequences (NLSs) and nuclear export sequences (NESs) [3,4,5].

During infection, some viruses integrate their genome into the host cell’s chromosomes, which can cause genetic alterations, mutagenesis, and cell death. In the case of retroviruses, such as human immunodeficiency virus (HIV), the integration of their genome into the host genome is a mandatory process for productive infection. Following infection, retroviruses generate linear, double-stranded cDNA through reverse transcription. This cDNA is flanked by long terminal repeats, which are part of a nucleoprotein complex known as the pre-integration complex. The pre-integration complex is composed of both viral and host cell proteins. The first step in the integration process is mediated by the retroviral integrase protein. When the pre-integration complex is located in the cytoplasm, the viral integrase hydrolyzes a dinucleotide from each 3 ’ end of the host cell’s DNA and catalyzes the asymmetric breakdown of this DNA to allow for the binding of viral DNA to this hydrolyzed end. The host cell’s DNA repair machinery subsequently cleaves the 5’ nucleotides that protrude from the viral DNA and fills those spaces with base pairs, stabilizing the viral genome insertion. Although all retroviruses can integrate cDNA into the host cell’s DNA, the integration can differ. Gammaretroviruses and foam retroviruses, for example, are integrated into sections of DNA, where they can bind to promoter regions and activate the transcription of genes. These sites are rich in CpG islands and are hypersensitive to DNase I activity. Conversely, lentiviruses prefer to integrate into sites where there are transcriptionally active genes. In the case of betaretroviruses, however, integration occurs randomly, and both alpharetrovirus and deltavirus integration sites are rich in CpG islands. In the case of DNA viruses, integration has not yet been fully described in all virus types. Here, we will refer to adeno-associated virus (AAV), about which a little more is known. In AAV, the proteins responsible for the integration of viral DNA into host DNA are mediated by replication (Rep) proteins, which are endonucleases. The integration step occurs at chromosome position 19q13.42 but may also involve integration at other sites [6,7]. Viral integration can cause multiple consequences for the virus and the host cell through uncontrolled proliferation or cell death. Additionally, viral integration can activate or silence transcription, which can result in viral latency [6].

DNA viruses replicate their genomes in the nucleus of host cells. To achieve this, they transport their genome into the nucleus through nuclear pore complexes (NPCs). However, RNA viruses can hijack cellular functions and localize their viral proteins to the nucleus–nucleolus, as is the case with the human immunodeficiency virus (HIV), where the host’s cytoplasmic proteins are recruited to the interior of the nucleolus to form and assemble viral ribonucleoprotein complexes [8]. Furthermore, research has shown that HIV uses the nucleolus as a site of immune evasion. Once HIV has synthesized its cDNA, it is transported into the nucleus, where the immune system of the host cell is unable to detect this external genetic material. In contrast, influenza A and hepatitis D viruses replicate and transcribe their viral genome in the nucleolus [9,10]. In this review, we discuss the crucial role of the nucleolus in the replicative cycle of different viruses.

## 2. Subnuclear Bodies and Their Importance during Viral Infections

Eukaryotic and prokaryotic cells differ in their compartmentalization of DNA. Eukaryotic cells contain DNA in an organelle called the nucleus, whereas prokaryotic cells package genetic material throughout the cytoplasm [11]. The nucleus is a dynamic structure composed of a double lipid membrane in which proteins give rise to nuclear pore complexes that connect the cytoplasm to the nucleus. Nuclear pores allow for the exchange of macromolecules between the nucleus and the cytoplasm. Therefore, they are considered important for gene regulation and cellular homeostasis, as they allow selective exchange between the nucleus and the cytoplasm. The first structures that delineate the inside of the nucleus from the cytoplasm are the nuclear membranes, which comprise the outer nuclear membrane (ONM) and the nuclear inner membrane (INM) [12]. The nucleus is made up of a highly viscous liquid that affects the diffusion rate of molecules into the nucleus, where DNA and proteins responsible for regulating the transcription, maintenance, and replication of the cellular genome are located. Additionally, smaller structures in the nucleus are not bound by membranes but consist of protein, RNA, and DNA aggregates. These aggregates are known as nuclear bodies and include Cajal bodies, promyelocytic leukemia bodies (PML bodies), and the nucleolus. These nuclear sub-compartments regulate transcription, splicing, ribosome biogenesis, and telomere maintenance [13].

### 2.1. Phosphatidylinositol Phosphates in the Formation of Nuclear Condensates

Cell compartmentalization makes biochemical reactions more efficient because the elements involved are more concentrated. However, there are cell concentrates that may or may not have a lipid membrane, but in the case of membrane compartments, it is well known that phosphatidylinositol phosphates (PIPs) play an important role. These PIPs participate in the dynamics of the cytoskeleton, vesicle trafficking, and signal transduction; however, their role within the cell is dependent on their location and state of phosphorylation. Generally, PIPs are evenly distributed within cell membranes, and the order they take in the membranes is specific to the type of membrane, which can define the interaction they will have with cellular proteins. Nuclear compartmentalization without a membrane can occur with the interaction of scaffolding proteins, nucleic acids, and lipids and is known to depend on the direct interaction of intrinsically disordered regions present in proteins, tandem regions present in nucleic acids, and hydrophobic acyl-chain lipids. In this case, nuclear PIPs (nPIPs) are present in compartments known as nuclear speckles. These nPIPs are important because they have been shown to interact with elements involved in the initiation and elongation of transcription [14,15,16].

One of the most important PIPs is phosphatidylinositol 4,5-bisphosphate (PIP2), which fulfills the function of remodeling chromatin, responding to DNA damage, and regulating gene expression. PIP2 is found in NSs, nucleoli, and the nucleoplasm. It is now known that NSs are associated with small nucleoproteins and the hyperphosphorylated form of RNA polymerase II. In the nucleolus, they can interact with proteins such as Fibrillarin and UBF, which participate in the transcription of ribosomal RNA (rRNA). Additionally, some reports have shown that PIP2 interacts with more than 300 nuclear proteins and participates in the splicing, pre-mRNA polyadenylation, rRNA processing, chromatin remodeling, histone acetylation/deacetylation, and regulation of the transcriptional activity of RNA polymers. These functions are achieved because of the ability of PIP2 to alter protein conformation and/or localization [17].

### 2.2. Cajal Bodies

Cajal bodies (CBs) are approximately 1 mm in diameter and exist at a density of between 1 and 6 per cell [18]. These bodies are not mobile; they are physically associated with specific active gene loci that are responsible for their formation. Their restricted “movement” is primarily regulated by the loop-out genomic regions that they are associated with and are located near sites of active transcription; however, CBs have predominantly been associated with the assembly and regeneration of transcription and splicing complexes [19]. Additionally, they participate in DNA repair and the processing and export of small nucleolar RNAs (snRNAs) [20,21]. The main proteins of CBs are p80-coilin, survival of motor neuron protein 1 (SMN-1), nucleolar phosphoprotein 140 (Nopp140), and WD40-encoding RNA antisense to p53 (WRAP53) [22]. Viruses known to interact with CB components include the Al minute virus of mice (MVM), African swine fever virus (ASFV), influenza A virus, Zika virus (ZIKV), and adenovirus 5 (Ad5) [23]. In MVM, subnuclear structures are formed during infection, in which the NS1 protein interacts with the SMN protein. This interaction aids the formation of subnuclear replication-associated bodies (APARs) that allow the virus to accumulate the necessary replication factors to complete its infection cycle [24].

During ASFV infection, CBs undergo remodeling, which is associated with the virus controlling the host cell’s transcription machinery [25]. During influenza A infection, the influenza nucleoprotein can interact with BCs, and the number of CBs per nucleus increases during infection [26]. This also occurs during ZIKV infection, where the NS5 protein interacts with the p80-coilin protein, increasing the number of CBs per nucleus [27]. During human Ad5 infection, the L4-22K protein can interact with the p80-coilin protein to facilitate the export of viral mRNA [28].

### 2.3. Promyelocytic Leukemia Bodies

Promyelocytic leukemia (PML) bodies are subnuclear structures associated with the regulation of apoptosis, transcription, DNA repair, and replication. Viruses associated with PML complex infections include human cytomegalovirus (HCMV), herpes simplex virus 1 (HSV-1), and adenovirus type 5. In the case of adenovirus type 5, PML bodies are remodeled during infection, and the E4-ORF3 protein interacts with the PML protein, which controls the DNA repair complex. This is achieved through interference with the DNA repair complex during infection, which prevents the formation of viral DNA aggregates that the immune system can recognize [29]. An interaction between the infected cell polypeptide protein (ICP0) and Sp100 occurs with HSV-1. This interaction causes the Sp100 protein to be eliminated. One of the functions of Sp100 is to serve as a sensor for DNA/protein complexes that are foreign to the host cell, and its elimination therefore allows for the successful implementation of the infection cycle [30]. It has been observed that the viral IE1 protein interacts with the PML protein during HMCV infection. This prevents SUMOylation, which is necessary for PML to activate an antiviral defense mechanism. In this way, HMCV counteracts the host’s defense mechanisms [31].

### 2.4. Paraspeckles

A new structure found in nuclear bodies, known as a paraspeckle, has recently been described. Paraspeckles are stress-induced structures containing approximately 40 RNA-binding proteins that can interact with a series of long non-coding RNAs (lncRNAs) to control gene expression. NEAT1, for example, is an abundant lncRNA in nuclear bodies and is of utmost importance in supporting the development of DNA under stress. This leads to cell proliferation; even in the presence of damage, this can lead to the development of cancer [32].

### 2.5. Nuclear Speckles

Nuclear speckles (NSs) are structures or domains that are found in nuclear bodies and are enriched in elements necessary for cell splicing. However, it has recently been shown that NSs can also participate in RNA metabolism. These highly dynamic structures move from the nuclear position when splicing is inhibited and return to their original sites when splicing is activated again. Although various elements are involved in the development and formation of NSs, the roles of the SON and SRRM2 proteins are unpredictable in their formation [33].

### 2.6. Super-Enhancers

Super-enhancers (SEs) are macromolecular condensates that contain elements that promote cellular gene expression. These condensates regulate gene expression by interacting with many transcription factors (TFs). Examples include Sox2, Nanog, and Klf4 [34].

### 2.7. The Nucleolus: Basic Composition and Principal Function

The nucleolus is a membraneless organelle that can be divided into three subcompartments: the fibrillar center (FC), dense fibrillar component (DFC), and granular component (GC). The primary function of the nucleolus is to participate in and coordinate the processing and biogenesis of pre-ribosome subunits and rRNAs [35]. These are the boundaries of the FC and DFC zones where the transcription of rDNA to rRNA occurs. Both rRNA and some nucleolar ribonucleoproteins (snoRNPs) accumulate in the DFC zone so that processing and post-transcriptional modification of the rRNA can be carried out. This results in the formation of 18S, 5.8S, and 28S rRNA. Finally, the late processing of rRNA occurs in the GC zone. In this zone, mature rRNA is processed by pre-ribosomal particles, and all subunits are transported to the cytoplasm. In this review, we provide a more detailed overview of the processes involved in the biogenesis of pre-ribosomal subunits [36].

#### 2.7.1. Transcription of rDNA

Transcription of rDNA is a crucial process in ribosome biogenesis in which rRNA, necessary for the formation of ribosomes, is synthesized. This process is initiated by the formation of a preinitiation complex in the promoter region of rDNA. This complex is composed of several key elements, including RNA polymerase I. This enzyme is responsible for rRNA synthesis. Unlike RNA polymerases II and III, which transcribe other types of RNA, RNA polymerase I is specifically involved in the transcription of genes encoding rRNA. Upstream binding factor (UBF) is a transcription factor that binds to the rDNA promoter and is essential for the formation of the preinitiation complex. Its main function is to stabilize DNA in a conformation that favors the binding of RNA polymerase I to other necessary TFs. Promoter selectivity factor (SL1), a multi-protein complex that recognizes and binds to the rDNA promoter in collaboration with the UBF, ensures that RNA polymerase I is correctly positioned at the transcription start site, facilitating the assembly of the pre-initiation complex [24,25].

Once this preinitiation complex has formed, RNA polymerase I begins to elongate the rRNA. During this phase, RNA polymerase I proceeds along the rDNA to synthesize a new strand of rRNA. Both UBF and SL1 remain associated with the transcription complex, thereby ensuring the stability and efficiency of the process. Elongation continues until the transcription release factor (PTRF), also known as Cavin-1, intervenes. This factor plays a crucial role in inducing the dissociation of RNA polymerase I from rDNA, allowing this enzyme to terminate the current transcript and initiate a new round of transcription. This mechanism is vital for the continuous and efficient production of ribosomes, which are essential for protein synthesis in cells. Dysfunctions in this process can lead to disorders in protein production and have been implicated in various human diseases, including some forms of cancer [26,27,28].

#### 2.7.2. Ribosomal RNA Processing

As an essential component of ribosomes, rRNA is the molecular machinery responsible for protein synthesis in all cells. During rDNA transcription, an rRNA precursor termed pre-rRNA is formed in eukaryotes, which contains 18s, 5.8s, and 28s rRNA sequences. This precursor is multigenic and includes several units that must be separated and modified to produce mature rRNAs [37,38].

Processing of pre-rRNA is a highly regulated and complex process that occurs primarily in the nucleolus, a substructure of the nucleus. This process involves several sequential steps and the participation of numerous molecules that act in concert to ensure rRNA maturation. The first step of pre-rRNA processing is excision, in which specific endonucleases cleave the pre-rRNA at specific sites to release individual rRNA precursors (18s, 5.8s, 28s). This cleavage is a highly controlled process that depends on the correct conformation of RNA and the intervention of several protein factors. As rRNAs are cleaved from their precursors, they undergo a series of post-transcriptional modifications crucial for their function in ribosomes. These modifications include the methylation of specific nucleotide residues and isomerization of uridines into pseudouridines. Methyltransferases and small molecule nucleolar RNAs (snoRNAs) guide these modifications, ensuring that they occur at the correct sites. During this process, pre-mRNAs undergo a series of structural changes facilitated by RNA chaperones, GTPases, and AAA-ATPases. These chaperones help rRNAs achieve the proper conformation essential for integration into ribosomal subunits. After the initial excision and post-transcriptional modifications, specific exonucleases trim the ends of the pre-rRNA, remove additional sequences, and refine the rRNAs to their final lengths. This step is crucial in ensuring that rRNAs correctly assemble into small (40s) and large (60s) ribosomal subunits [39,40,41].

#### 2.7.3. Subunit Assembly

The assembly of ribosomal subunits is essential for protein synthesis in cells. This process occurs in the nucleolus, where rRNAs are transcribed and modified. The smaller subunit (40s and 18s rRNA) is associated with approximately 33 ribosomal proteins. In the large (60s) subunit, the 28s, 5s, 8s, and 5s rRNAs assemble into approximately 49 ribosomal proteins. Once these ribosomal subunits are assembled, they are exported to the cytoplasm through the nuclear pores. In the cytoplasm, the 40s and 60s subunits combine to form complete and functional ribosomes that are essential for the translation of mRNA into proteins. This assembly process is highly regulated and crucial for cellular homeostasis, as ribosomes are responsible for the synthesis of all cellular proteins [30].

#### 2.7.4. The Nucleolus as a Multifaceted Nuclear Compartment

Although the nucleolus is a structure that is known for its role in the biogenesis of rRNAs and ribosomes, studies have reported that it is also responsible for gene silencing, cell cycle regulation, the cellular stress response, protein control, telomerase assembly, editing of other RNAs, cell cycle progression, DNA repair, and immune system activation [42,43,44,45,46]. Indeed, the nucleolus is involved in additional cellular functions that may not be related to the biogenesis of ribosomal subunits, largely because more than 4500 proteins are associated with the nucleolus at different cellular stages. Additionally, the protein content of the nucleolus has been shown to be dynamic and is altered in response to cellular stress. Three proteins are associated with the nucleolus: nucleolin, fibrillarin, and B23/Nucleophosmin [47] (Table 1). We describe these proteins in more detail below.

B23: The nucleophosmin protein participates in DNA damage repair, genomic stability, tumorigenesis, ribosome assembly, nucleocytoplasmic transport, and regulation of rDNA transcription by generating changes in the chromatin structure [53]. The B23 protein participates by binding to nucleic acids and pre-ribosomal particles to allow the biogenesis of ribosomes and is primarily found in the nucleolus but can also be found in the cytoplasm [54,55]. Additionally, B23 has been shown to have ribonuclease activity and is capable of preribosomal RNA processing to form mature rRNAs. Furthermore, B23 inhibits the formation of protein aggregates within the nucleolus to form ribosomal subunits [55,56].

Fibrillarin: The fibrillarin (FBL) protein is a 2′-O-methyltransferase that catalyzes RNA base modifications on rRNA under the guidance of BOX C/D snoRNAs that is conserved in both Archaea and Eukarya [57]. This protein catalyzes the methylation of rRNA. The basic structure of FBL consists of four main domains: a spacer region, RNA-binding domain, methyltransferase domain, and alpha region [58]. In eukaryotic cells, FBL is predominantly associated with proteins such as Nop56 and Nop58 and some snoRNAs [49,59]. Together, these proteins and snoRNAs form what are known as ribonucleoprotein complexes. Found in the nucleoli and CBs of cells, FBL has been used as a nucleoli marker. In addition to participating in the biogenesis of rRNAs, FBL acts as a transcriptional regulator, cellular stress sensor, and ribonuclease [60].

Nucleolin: Nucleolin is an abundant non-ribosomal protein located in the nucleolus. In addition to participating in the metabolism of DNA and RNA, nucleolin participates as a histone chaperone, modulating chromatin remodeling and DNA repair and replication and participating in the transcription, splicing, and transport of mRNA [61,62,63]. Nucleolin is also involved in the regulation of mRNA by binding to its 3’UTR region, which suppresses translation. It also interacts with the spliceosome to regulate the splicing and transcription of RNA polymer I [63].

Nop56: Nucleolar protein 56 (Nop56) is a 66 kDa nucleolar protein that participates in the processing of rRNAs. It is the last protein to bind to the snoRNA c/d Box complex, and it is necessary for this protein to be SUMOylated so that it can be located in the nucleolus and bind to snoRNAs [64].

Non-protein nucleolus components: In addition to the proteins necessary for the processing and assembly of rRNAs, small ribonucleolar protein particles (snoRNPs) are also present. These complexes directly modify and participate in the maturation of rRNAs [65]. The elemental composition of snoRNPs includes small nucleolar RNAs (snoRNAs) of approximately 60–300 nt in length that are associated with nucleolar proteins [66]. However, the most abundant snoRNP in the nucleolus is the snoRNP C/D Box, composed of a snoRNA with a C box (RUGAUGA) and a D box (CUGA) [66,67,68]. This complex is associated with four proteins: Nop56, nucleolar protein 58 (Nop58), small nuclear protein 13 (SNU13), and fibrillarin protein [52,59,69,70,71]. The assembly of the RNP C/D box complex begins by binding RNAs to SNU13, which forms a rotating structure that promotes the formation of C/D box motifs. Subsequently, Nop58 binds to this complex to form a heteromer, which allows the snoRNP complex to be blocked and to remain static so that the fibrillarin protein can methylate rRNAs. Finally, Nop56 binds in a similar way to Nop58 and can form heterodimers that stabilize the rRNA [72].

During viral infections, viruses modify the nucleus, generating an alteration in the transcription of rDNA and the biogenesis of ribosomes. This impacts the level of synthesis protein. In addition, inhibiting the formation of ribosomes can save resources and energy, which can be used for viral replication [73]. Viruses can occupy the nucleolus as a site for the transcription and maturation of viral RNAs and serve as a site for the assembly of viral particles [74,75,76].

The nucleolus is used for various processes in viral infections, serving as a site of transcription and maturation of viral RNA and the assembly of viral particles [77]. Additionally, the nucleolus contains nucleolar proteins that can be used during viral infections to form pro-viral microenvironments, as is the case with viroplasms [76].

## 3. Involvement of Nucleolus Components during Viral Infections

### 3.1. Viral Proteins and Their Interaction with Nucleolus Components

Although RNA viruses generally carry out their entire infection cycle in the cytoplasm, some viruses access or import their proteins into the nucleolus to interact with resident proteins (Figure 1) [78]. The interaction of nucleolar and viral proteins is a process that allows viruses to replicate, transcribe, and, in some cases, assemble and transport their viral particles. When viral proteins enter the nucleolus, they compete with the resident proteins of the nucleolus, which generates an imbalance of the existing structures and the functions that these structures carry out in a “normal” way. This leads to the alteration of processes, such as ribosome biogenesis, cell cycle, and cell death. Interactions between viral and nucleolar proteins will be described in the following section [78,79,80,81].

During HIV infection, the HIV Rev protein facilitates the nuclear export of intron-containing HIV mRNAs [82]. Once in the nucleolus, these are retained for HIV to regulate the transcription of viral genes [83,84]. In addition, the Tat protein can interact with the FBL protein and snoRNA U3 to interfere with ribosome biogenesis [79]. In contrast, eliminating the FBL protein inhibits the production of viral particles in Hendra and Nipa viruses. Microscopy and co-immunoprecipitation assays have shown that viral matrix proteins interact with FBL, causing a change in their distribution to regulate the host’s proviral genes. This improves the synthesis of viral proteins by improving the methylation of viral RNAs [85].

Non-structural protein 1 (NS1) has a nucleolar localization domain in the influenza A virus, where it interacts with nucleolar proteins such as nucleolin, B23, and FBL. The latter alters the binding capacity of these proteins to rDNA and induces hypermethylation. This leads to the activation of nucleolar cellular stress pathways and stops rRNA processing [86,87,88].

Another virus that interacts with the cell nucleolus is the Epstein–Barr virus, which encodes a snoRNA, called v-snoRNA1, during its infection cycle. This snoRNA interacts with the nucleolar proteins NOP56, NOP58, and FBL, as well as the nucleolar snoRNA U3. This interaction affects rRNA maturation and protein synthesis. However, further studies are needed to understand how this mechanism promotes infection [89,90,91].

### 3.2. Modulation of Host Genes and rDNA Transcription during Viral Infections

During cellular stress, changes occur in the transcriptional regulation of genomic DNA and rDNA. The nucleolus is a crucial sensor that coordinates the cellular stress responses and transcription. Under stressful conditions, ribosomal biogenesis stops as the cell attempts to save resources and energy. Under normal conditions, ribosome biogenesis is directly related to cellular needs. However, during viral infection, transcription changes, and the increase in production of ribosomes, which stimulates protein synthesis, is generally promoted [4,92]. For example, in the hepatitis C virus, NS5A proteins can associate with the nucleolar protein UBF to induce hyperphosphorylation, thus increasing the transcription of ribosomal genes. However, ribosome biogenesis is suppressed in some cases. Viral proteins typically target RNA polymerase I to modulate transcription. In contrast, a mechanism observed during viral infections is the activation of proteins related to the detection and response to DNA damage, which stops the transcription of rRNA [93].

(A) Viruses that have an interaction with elements of the nucleolus: The HIV TAT protein can interact with fibrillarin in the nucleolus, altering ribosome biogenesis and protein production. The NS1 protein of the influenza virus interferes with nucleolin, inhibiting interferon production and facilitating viral replication. Finally, v-snoRNA1 from the Epstein–Barr virus alters the function of NOP56 and NOP58 in the nucleolus, helping the virus to evade the host immune response. (B) Viruses that have interactions with elements of Cajal bodies: Cajal bodies are essential in ribonucleoprotein biogenesis and RNA processing. The P80-coilin protein, key to the organization of these bodies, may interact with the Zika virus NS5 protein, perturbing its function and affecting critical nuclear processes such as RNA splicing. This could influence viral replication by altering cellular factors. In addition, the p80-coilin protein also interacts with the L4-22K protein of adenovirus 5, modifying its distribution and facilitating viral messenger RNA production. In MVM infections, the SMN-1 protein can be sequestered at sites of viral replication, suggesting viral manipulation of the host machinery to favor replication. (C) Viruses that have capacity to interact with leukemia proteolytic bodies: PML subnuclear structures play an important antiviral role since, in some cases, they can inhibit viral replication by sequestering DNA or some viral proteins, as in the case of HIV infections. In the case of adenovirus, the E4-ORF3 protein reorganizes PML nuclear bodies, interfering with their antiviral functions and helping the adenovirus to evade host response. Herpes simplex type 1 ICP0 protein and Sp100 protein interact to degrade Sp100 and disperse PML bodies, facilitating viral replication. Similarly, human cytomegalovirus IE1 protein disorganizes PML bodies to evade antiviral responses and favor viral replication. Another important aspect is that they can modulate the innate immune response by promoting the activation of type I interferons through transcription factors such as RF3 and IRF7 crucial in antiviral activity.

## 4. The Role of Non-Coding RNAs in Viral Infections

Research has focused on mRNA, rRNAs, and transferred RNAs. However, it has been shown that more than 90% of the genome corresponds to non-coding RNAs (ncRNAs). These ncRNAs can be divided into two groups: those that are abundantly expressed and regulate essential cellular functions, and those that regulate gene expression at the epigenetic, transcriptional, and post-transcriptional levels. The first group comprises ribosomal, nuclear, transfer, and small nucleolar RNAs (snoRNAs). The second group comprises small non-coding RNAs (sncRNAs) and large non-coding RNAs (lncRNA) [94]. SncRNAs include miRNAs, siRNAs, and small piwi-interacting RNAs (piRNAs). The main difference between sncRNAs and lncRNAs is that the latter have been proposed to be regulators of miRNAs, piRNAs, and snoRNAs [95,96].

During viral infections, host cells begin to express lncRNAs abnormally, mainly affecting cellular immune and antiviral responses (Figure 2). Most lncRNAs involved in these infections influence the secretion of interferons and cytokines, affecting the expression of interferon-stimulated genes and interfering with pattern recognition signaling via cellular receptors [97,98].

In Kaposi’s sarcoma-associated herpes virus (KSHV) infection, the lncRNA PAN is generated, which activates viral replication and regulates host cell gene expression. This is because PAN can bind to the JMJD3 and UTX demethylase promoters necessary to initiate viral replication [99,100,101]. During infection with yellow fever virus (YFV) and West Nile virus (WNV), lncRNA sfRNA is produced, which is essential for replication of both viruses. However, such lncRNA represses viral replication in the Japanese encephalitis virus (JEV) [98,102,103,104]. In the case of Epstein–Barr virus, lncRNA BHLF1 is synthesized, which promotes viral replication [105]. During HCMV infection, lncRNA2.7 is expressed in the first hours of infection to interact with complex I of the electron chain and maintain stable mitochondrial membrane potential, thus producing the ATP necessary for the virus [106].

Another mechanism by which viruses can improve their infection cycle is through the regulation of the host cell’s gene expression through the actions of lncRNAs at the transcriptional level. For example, lncRNA PAN can bind to the latency-associated nuclear antigen during KSHV infection, which is involved in chromatin remodeling. This binding decreases histone trimethylation and therefore prevents gene expression. Furthermore, PAN binds to the TF known as IRF4 to inhibit gene expression. Another mechanism by which lncRNAs generated during viral infections regulate gene expression is by altering the degradation of mRNAs [99,101,107].

Once the viral genomes and proteins have been synthesized, they must be assembled and released to infect other cells. It is now known that lncRNAs can participate in these processes. For example, the BocaSR lncRNA of human Bocavirus 1 (HBoV1) is essential for the viral replication cycle and participates in the packaging and formation of mature viral particles [108]. In contrast, the JEV lncRNA known as sfRNA, which is present during the late stages of the infection cycle, has been reported to promote the packaging and release of viral particles [102,104].

Furthermore, lncRNAs may be involved in the evasion of the immune system during viral infections. In the case of KSHV, lncRNA PAN inhibits antiviral factors such as IL-4, IL-8, IFN-16, and IFN-γ by binding to TFs such as PU.1, H1/H2A, and SSBP [109]. In WNV infection, sfRNAs have been reported to inhibit apoptosis and decrease IFN production by influencing the translocation and activation of IRF-3 [103].

## 5. Therapeutic Targets for Viral Replication: Nucleolus and Cytoplasm–Nucleus Traffic

Nucleus–cytoplasm trafficking regulates various cellular functions, including protein synthesis and processing, RNA transcription, and the cellular stress response. This trafficking is critical during viral infections, when many viruses use cellular transport mechanisms to replicate and spread efficiently [110]. The nucleolus is the largest organelle in the nucleus and plays a crucial role in ribosome biogenesis and other essential cellular functions. The nucleolus is responsible for ribosome production and is involved in cell cycle regulation, stress responses, DNA repair, and immune system activation. During viral infections, viruses often sequester components of the nucleolus to facilitate their replication and assembly. This sequestration can disrupt normal nucleolar functions and allow viruses to use cellular resources to their advantage [111].

Altering these processes may be an effective strategy for combating viral infections. Various anticancer and antiviral drugs have been shown to interrupt nuclear–cytoplasmic trafficking, inhibit viral replication, and block access to the nucleolus (Figure 3). These drugs act on critical components of nucleolar trafficking and function, inducing nucleolar stress, inhibiting rRNA synthesis, and disorganizing the nucleolar structure [112]. By understanding these mechanisms, more effective and targeted therapeutic strategies can be developed (Figure 3). Table 2 summarizes several drugs that affect nuclear–cytoplasmic and nucleolar trafficking, detailing their mechanisms of action and specific targets (Table 2).

### 5.1. Mechanism of Action of Drugs

#### 5.1.1. BMH-21

The main mechanism of action of BMH-21 is the inhibition of rRNA transcription. This is achieved by its binding to the promoter region of the 47S ribosomal gene, which blocks the function of RNA polymerase I. As a result, proper ribosome formation is prevented, and nucleolar stress is induced. Although BMH-21 has shown potential in cancer research, its use in clinical studies has been limited due to excessive DNA damage. However, when doses are adjusted, BMH-21 could be a promising candidate for antiviral therapy against certain viruses that depend on host RNA polymerase I to transcribe their genomes and synthesize mRNAs necessary for viral protein production, such as HIV and some viruses belonging to the Arenaviridae family [113].

#### 5.1.2. CDK Inhibitors

The use of CDK inhibitors alters the formation and maintenance of nucleoli after mitosis. These inhibitors have been reported to interfere with rDNA transcription, attenuating pre-RNA transcription. Indeed, the use of CDK inhibitors also alters the processing of ribosomal pre-RNA. With regard to viral infections, the T antigen of the SV-40 virus has been shown to bind to the rDNA promoter to activate rRNA transcription. In contrast, the hepatitis C virus can positively regulate the transcription of rDNA through the action of the NS5A protein. Another virus that has been reported to interact with rDNA through its E6 and E7 proteins is human papillomavirus 16. However, the biological relevance of the interaction of these viruses with rDNA is not yet clear; it has been hypothesized that they function by affecting the biogenesis of ribosomes, synthesis of proteins, or by activating antiviral mechanisms. Although there are still no studies discussing the use of CDK inhibitors as an antiviral treatment, it is considered a potential antiviral strategy [93,119,120,121].

#### 5.1.3. Cisplatin

Cisplatin is a chemotherapeutic drug that is primarily involved in interacting with DNA. In viral infections, cisplatin has been used as an antiviral agent because of its ability to block DNA replication and transcription by generating platinum DNA adducts. Currently, the use of cisplatin against viruses such as herpes simplex virus type 1, influenza A, influenza B, papovavirus SV40, Newcastle disease, vesicular stomatitis virus, and adenoviruses 4 and 5 has been described. Although the mechanism of action is not yet fully understood, cisplatin is believed to act by interacting with TFs, inhibiting enzymes (cellular and viral), interfering with the cell cycle, and interfering with the infection cycle [122].

#### 5.1.4. 5-Fluorouracil (5-FU)

Various viruses have been reported to make use of the resident elements of the nucleolus to carry out a successful infection cycle. In recent years, attempts have been made to develop an antiviral approach to attack the nucleolus during viral infections. The use of 5-fluorouracil (5-FU), an analog of uracil with a fluorine atom at the C-5 position instead of hydrogen, as an antiviral agent has been proposed. One of the proposed mechanisms by which 5-U could have an antiviral effect is that it can alter nucleolar structures, generating a change in the organization and function of the resident proteins, which could affect viral infections. This is because they decrease the availability of the necessary elements needed for a successful replication cycle. Currently, 5-FU is used in combination with deoxyribonucleosides and deoxyribose as a possible treatment for SARS-CoV-2 infection. However, this hypothesis remains to be tested [123].

#### 5.1.5. Camptothecin

Camptothecin (CPT) affects the nucleolus by inhibiting topoisomerase I, an enzyme crucial for the unwinding of DNA during rRNA transcription. This inhibition interferes with ribosome production, leading to disruption of the nucleolus and activation of cellular stress response pathways, such as apoptosis. These effects contribute to the cytotoxicity observed in CPT-treated cells. In addition to its effects on the host cell, CPT has also been studied for its impact on the replication of various viruses. In the case of avian adenovirus serotype 4, CPT inhibits infection by interfering with viral DNA synthesis, causing DNA fragmentation and blocking viral replication. In enterovirus 71, CPT inhibits both viral RNA replication and protein synthesis by blocking DNA topoisomerase I, thereby preventing viral propagation within the host cell. However, CPT is not equally effective against all viruses. In measles virus infection, CPT does not show a significant antiviral effect on its own, although it may be effective when used in combination with oncolytic therapies, suggesting its potential use in combination treatments to enhance its efficacy in certain contexts [124,125,126].

#### 5.1.6. Doxorubicin

Doxorubicin is a widely used chemotherapeutic agent that predominantly acts through the inhibition of topoisomerase II. This causes DNA breaks, interferes with replication and transcription, and promotes cancer cell apoptosis. Doxorubicin can also induce alterations in the structure of the nucleolus, which manifests as fragmentation or reorganization. These changes may contribute to cellular dysfunction and facilitate apoptosis.

A derivative of doxorubicin has recently been developed that has been modified to act as an antiviral in addition to its known use as a chemotherapeutic agent. This new derivative has been shown to inhibit the replication of Dengue and yellow fever viruses in vitro. In addition, evidence suggests that this derivative can influence the viral protease of SARS-CoV-2, the virus causing COVID-19. As this viral protease is crucial for SARS-CoV-2 replication, its inhibition could reduce the ability of the virus to replicate and cause disease [127,128].

#### 5.1.7. Atorvastatin and Ivermectin

Several viruses are known to occupy the alpha/beta 1 import pathway to transport their proteins into the nucleus, enhance their replication, and prevent activation of the cellular immune response. Examples of viruses that use this pathway include Zika, West Nile, Japanese encephalitis, hepatitis C, and Dengue viruses. These viruses carry viral proteins, such as viral polymerase NS5, viral protease NS3, and capsid proteins (C). Therefore, the inhibition of cellular nuclear transport is currently an antiviral strategy. In this regard, both ivermectin and atorvastatin have been shown to affect the import of elements into the nucleus, whereby they promote the cytoplasmic accumulation of alpha importins, which leads to a reduction in infection. This is because these elements are necessary to carry the replication cycle in the nucleus [116].

#### 5.1.8. Leptomycin B

Maintenance of the chromosomal region (CRM1), also known as exportine 1 (XPO1), plays a key role in the replication of viruses from different families, including retroviruses, ortomyoscoviruses, paramyxoviruses, flaviviruses, coronaviruses, rhabdoviruses, and herpesviruses. This is the main recipient of nuclear protein and RNA export and contains a nuclear export signal (NES). Both HIV-1 and HTLV-1 use CRM1 to mobilize viral mRNA complexes from the nucleus to the cytoplasm with the proteins Rev (in the case of HIV-1) and Rex (in the case of HTLV-1). Conversely, Leptomycin B (LMB) is an antifungal and antibiotic that acts by blocking nuclear export by binding to CRM1 in Cys528, which resides in its NES junction slot and inhibits the binding of the charge to CRM1. The interruption of exports mediated by CRM1 causes changes in virion protein expression, virion replication, incomplete viral assembly, reduced infectivity, and improved host antiviral immune responses due to CRM1-mediated export disruption [129,130].

#### 5.1.9. Selinexor

Viruses hijack the cellular import and export machinery to enable the transport of proteins from the host cell to the nucleolus so that they cannot exert antiviral activity. Indeed, the import of proteins from the cytoplasm to the nucleus is regulated by importins that recognize a nuclear import sequence (NLS) present in the protein to be imported. In contrast, import from the nucleus into the cytoplasm is mediated by XPO1, a protein that recognizes NES. Therefore, the inhibition of some import/export elements could be used as an antiviral therapy. In this regard, there are selective nuclear export inhibitors (SINEs) that have shown good antiviral activity; however, here, we will focus specifically on the selinexor inhibitor, which binds to XPO1 and blocks the transport of molecules from the nucleus to the cytoplasm. Inhibitors of XPO1 have been shown to have antiviral capacity in various viruses, such as influenza virus, respiratory syncytial virus, HIV, and SARS-CoV-2 [131,132,133].

#### 5.1.10. Quarfloxin (CX-3543)

Non-canonical nucleic acid structures are formed during replication/transcription and are known as G-quadruplexes (G4). These structures form G-rich DNA or RNA sequences, and their formation has been described in viruses such as herpes simplex types 1 and 2, varicella zoster, human cytomegalovirus, Epstein–Barr, Kaposi’s sarcoma herpes, SARS-Cov-2, hepatitis C, Zika, West Nile, Chikungu-nya, and HIV-1 viruses. Additionally, the formation of viral G4s modulates viral replication and transcription in most viruses, and in some cases, viral proteins are capable of specifically recognizing viral and host G5 regions. Therefore, drugs targeting viral G4 have been developed. There are more than 2800 small molecules that recognize G4, but we will focus on CX-3543, also known as quarfloxin, in this review.

CX-3543 has an antiviral effect because the viral genome is more accessible to CX-3543 while the genome of the host cell is highly condensed; therefore, its binding means that the viral genome is no longer accessible for replication or interaction with viral proteins that are responsible for the formation of mature viral particles [134,135,136,137,138,139,140,141,142,143].

## 6. Conclusions

Viruses have developed various strategies to maintain active and effective infection over time. As previously mentioned, the nucleolus is a crucial site for the regulation of many cellular functions, mainly due to its proteins. Some viral proteins interact with nucleolar components to allow for efficient replication or transcription and may also be involved in the assembly and transport of viral genetic material from the nucleoplasm to the cytoplasm. Furthermore, viral proteins entering the nucleolus can interact with non-protein elements such as ncRNAs, suggesting the need for additional strategies to regulate their survival during the infection cycle. However, several of these mechanisms remain unclear. Studying and understanding the interaction between viruses and the nucleolus are essential for developing new antiviral strategies.

## Figures and Tables

**Figure 1 cells-13-01591-f001:**
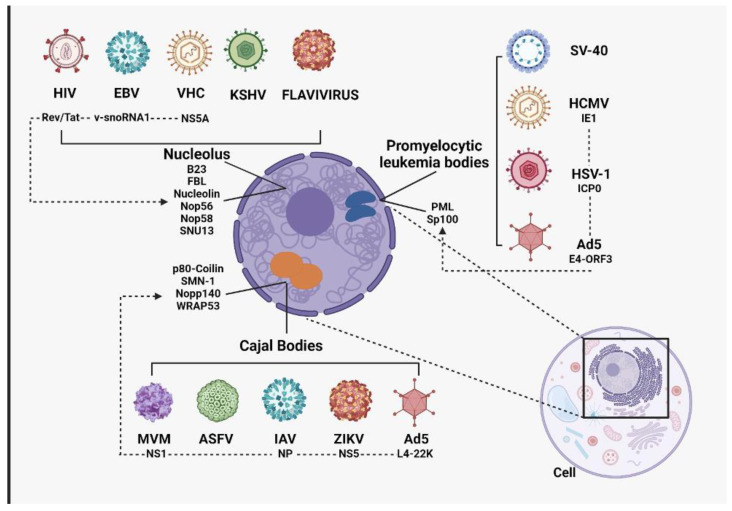
Subnuclear structures and their interaction with viruses.

**Figure 2 cells-13-01591-f002:**
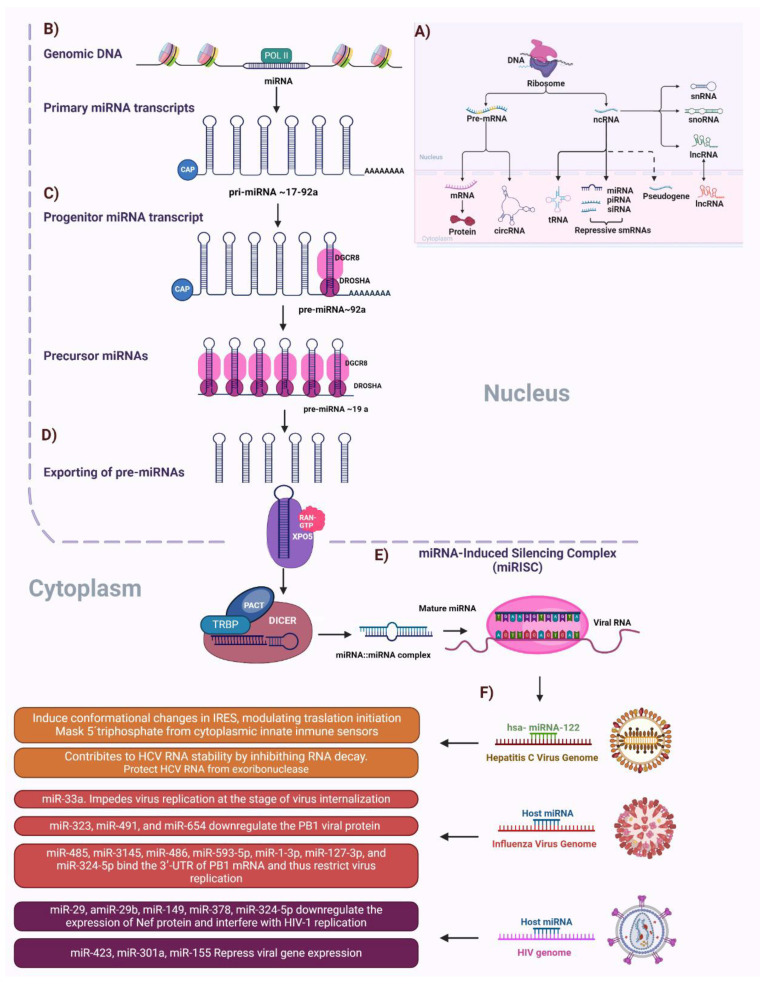
The ole of non-coding RNAs during viral infections. (**A**) RNA molecules play various roles in cells, and they are broadly classified into two categories: coding RNA and non-coding RNA. Non-coding RNAs do not code for proteins but have crucial roles in regulating gene expression and maintaining cellular functions. Here is an overview of the main types of ncRNA: microRNA (miRNA), small interfering RNA (siRNA), piwi-Interacting RNA (piRNA), small nuclear RNA (snRNA), small nucleolar RNA (snoRNA), long non-coding RNA (lncRNA), circular RNA (circRNA), and long intergenic non-coding RNA (lincRNA). (**B**) miRNAs are small, non-coding, single-stranded RNAs ~23 nt (ranging from 19 to 25 nt). The majority of mammalian miRNAs genes are located in intergenic regions or in antisense orientation and are transcribed by RNA polymerase II (Pol II) as primary miRNA transcripts (pri-miRNAs). (**C**) pri-miRNAs are capped, polyadenylated, and contain a local stem–loop structure that encodes miRNA sequences in the arm of the stem. This stem–loop structure is cleaved by the nuclear RNase III type enzyme Drosha in a process known as ‘cropping’. In the nucleus, the RNA hairpin structure is excised by the RNAse III-like enzyme Drosha and its co-factor DGCR8 to form the precursor miRNA (pre-miRNA). (**D**) pre-miRNA is translocated to the cytosol by exportin5, where it is processed by the Dicer protein complex, resulting in an miRNA duplex (miRNA/miRNA*), which is made up of a guide chain (miRNA) and a passenger chain (miR-NA*). (**E**) The miRNA/miRNA* is then loaded into the Argonaute (AGO), promoting the expulsion and degradation of the miRNA and the formation of the RNA-induced silencing complex (RISC). The RISC recognizes the targeted mRNA through base-pairing with miRNA. (**F**) miRNAs function as key regulators of gene expression in many different cellular pathways and systems, including immune response. So, several viruses with the purpose of carrying out an efficient replication or a persistent infection are able to modify their biogenesis, such as in the case of HIV. In this sense, several studies report an increase in miRNAs that facilitate its replication while inhibiting the Dicer–TRBP–PACT complex.

**Figure 3 cells-13-01591-f003:**
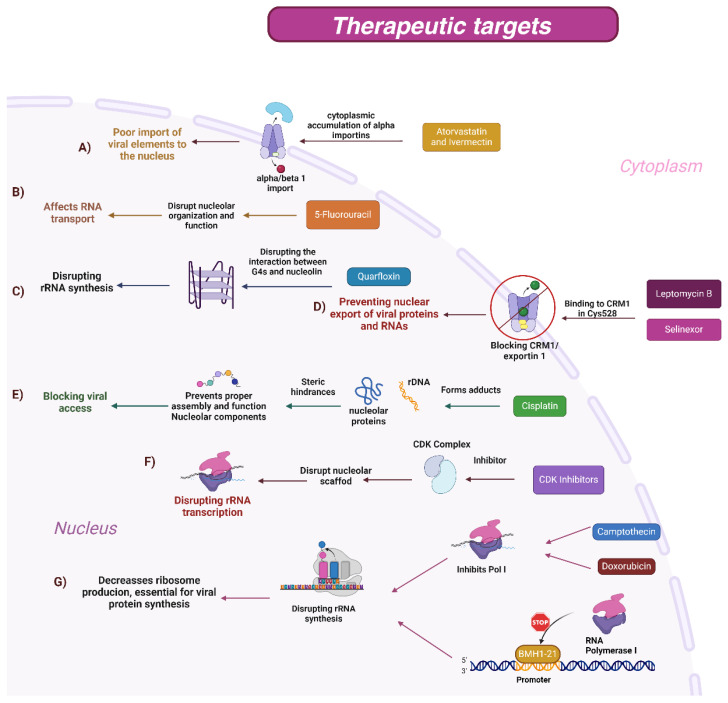
(**A**) Atorvastatin and Ivermectin: Both drugs are known to disrupt the nuclear–cytoplasmic transport of proteins. Specifically, they impair the trafficking of viral proteins, which is crucial for the assembly and maturation of viral particles. This action has been noted in viruses like Dengue and Zika virus (ZIKV), where effective viral replication relies on the proper localization of viral proteins within the host cell. (**B**) 5-Fluorouracil (5-FU): A pyrimidine analog that interferes with nucleotide metabolism and RNA function. 5-FU targets nucleolar structures, disrupting the organization and function of the nucleolus. This disruption impairs the transport and processing of rRNA and other molecules, crucial for ribosome assembly and function. (**C**) Quarfloxin (CX-3543): Its main mechanism is the inhibition of the interaction between the nucleolar protein nucleophosmin (NPM1) and DNA containing regions rich in G-quadruplexes, secondary structures present in promoter regions of ribosomal DNA. This drug destabilizes ribosome assembly by blocking the transcription of ribosomal RNA, which reduces protein production in the cell. (**D**) Leptomycin B: Inhibits CRM1 (also known as exportin 1), a key protein in the nuclear export of proteins and RNAs. By blocking the nuclear export of viral proteins and RNAs, Leptomycin B effectively prevents the replication of various viruses, including HIV and Influenza. This inhibition disrupts the life cycle of these viruses, which rely on the export of viral components for replication and assembly. Selinexor: Another inhibitor of CRM1/exportin 1, like Leptomycin B. Used in the treatment of certain cancers and viral infections, Selinexor blocks the nuclear export of viral and cellular components, thereby disrupting viral replication and cancer cell proliferation by affecting cellular stress responses and apoptotic pathways. (**E**) Cisplatin: Forms covalent adducts with DNA, including ribosomal DNA (rDNA), and proteins within the nucleolus. These adducts create steric hindrances that prevent the proper assembly and function of nucleolar components. This action blocks the synthesis and maturation of rRNA, thereby hindering viral access to the nucleolar machinery necessary for replication. (**F**) CDK inhibitors: Target cyclin-dependent kinases (CDKs), which are critical regulators of cell cycle progression and nucleolar function. These inhibitors disrupt the nucleolar scaffold, leading to nucleolar dissolution. This disruption affects rRNA transcription and processing, impairing the nucleolus’s ability to produce ribosomes, which are necessary for protein synthesis, including viral proteins. (**G**) Camptothecin and Doxorubicin: Inhibit RNA polymerase I (Pol I), which is responsible for the transcription of rRNA genes. These drugs reduce the synthesis of rRNA, leading to decreased ribosome production. Since ribosomes are essential for the translation of viral proteins, their reduced availability impairs viral replication. BMH-21: Exerts its action by binding to DNA in rRNA gene regions, which leads to inhibition of RNA polymerase I and degradation of the enzyme. This inhibition specifically affects cells with a high rate of rRNA synthesis, such as tumor cells, without severely impacting normal cells.

**Table 1 cells-13-01591-t001:** Functional characteristics of nucleolar proteins.

Nucleolar Protein	Function	References
B23 (Nucleophosmin)	Participates in ribosome assembly, nucleocytoplasmic transport, and regulation of ribosomal DNA transcription. Has ribonuclease activity and prevents the formation of protein aggregates. Considered a core protein of the C/D box small nucleolar ribonucleoprotein (snoRNP) particles.	[48]
Fibrillarin (FBL)	S-adenosylmethionine-dependent methyltransferase that catalyzes ribosomal RNA methylation. Acts as a transcriptional regulator, cellular stress sensor, and ribonuclease. Considered a core protein of the C/D box small nucleolar ribonucleoprotein (snoRNP) particles.	[49]
Nucleolin	Involved in DNA and RNA metabolism, chromatin remodeling, DNA repair and replication, transcription, splicing, and transport of messenger RNAs.	[50]
Nop56	Participates in ribosomal RNA processing and the formation of small nucleolar ribonucleoprotein complexes (snoRNPs). Necessary for SUMOylation and nucleolar localization. Considered a core protein of the C/D box small nucleolar ribonucleoprotein (snoRNP) particles.	[51]
Nop58	Forms heteromers with Nop56 to stabilize ribosomal RNA in the C/D Box snoRNP complex, allowing ribosomal RNA methylation by fibrillarin. Considered a core protein of the C/D box small nucleolar ribonucleoprotein (snoRNP) particles	[51]
SNU13	Binds to RNA to form rotational structures, promoting the formation of C/D box motifs and facilitating ribosomal RNA methylation. Considered a core protein of the C/D box small nucleolar ribonucleoprotein (snoRNP) particles.	[52]

**Table 2 cells-13-01591-t002:** Drugs with potential antiviral activity and their effect regulating function in the nucleus.

Drug	Mechanism of Action	Target and Effects	References
BMH-21	Inhibition of rRNA transcription.	Decreases RNAm production, essential for viral protein synthesis.	[113]
CDK Inhibitors	Disrupt nucleolar scaffold, causing nucleolar dissolution and affecting rRNA transcription.	Impairs nucleolar function and nucleus–cytoplasm trafficking.	[114]
Cisplatin	Forms adducts with rDNA and nucleolar proteins, creating steric hindrances.	Prevents proper assembly and function of nucleolar components, blocking viral access.	[115]
5-Fluorouracil (5-FU)	Targets nucleolar structures, disrupting nucleolar organization and function.	Affects the transport of rRNA and other molecules.	[115]
Camptothecin	Inhibits Pol I, reducing rRNA synthesis.	Decreases ribosome production, essential for viral protein synthesis.	[114]
Doxorubicin	Inhibits Pol I, reducing rRNA synthesis.	Decreases ribosome production, essential for viral protein synthesis.	[114]
Atorvastatin and Ivermectin	Disrupts nuclear–cytoplasmic transport of viral proteins.	Impairs trafficking in Dengue virus and ZIKV.	[116]
Leptomycin B	Inhibits CRM1/exportin 1, preventing nuclear export of viral proteins and RNAs.	Blocks replication of HIV, Influenza, and other viruses.	[116]
Selinexor	Inhibits CRM1/exportin 1, blocking nuclear export of viral components.	Used in cancers and viral infections.	[117]
Quarfloxin (CX-3543)	Inhibits RNA polymerase I (Pol I), reduces ribosomal RNA (rRNA) synthesis.	Disrupts the nucleolin–rDNA complex	[118]

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
