# Peer review of "The Nucleolus and Its Interactions with Viral Proteins Required for Successful Infection"

_cells, 2024, doi:10.3390/cells13181591_

Round 1
Reviewer 1 Report
Comments and Suggestions for Authors
In this review, the authors discuss the general role of nuclear bodies, the nucleolus, and others in the cell nucleus's structural organization and gene expression. Later, they focus on the specific roles of nuclear compartments to promote viral replication. The review is not entirely well structured and requires some additional editorial work before publication. The figures are detailed and informative. Authors should also cite some recent papers that have been suggested.
Specific criticisms are outlined below:
Lane 36: “The canonical function of nuclear bodies is usually associated with RNA processing and ribosome biogenesis.” This statement is not entirely true and must be corrected. In principle, nuclear bodies, in general, regulate the concentration, exclusion, sequestration, assembly, modification, and recycling of specific components to regulate specific reactions involved in ribosome biogenesis, RNA transcription, and RNA processing.
Lane 66: The authors should highlight the significance of the integration of the viral genome(s) into the genome of host cells. They should discuss the generation of a double-strand break on the cellular chromosome, specific positions of the integration sites, and the virus latency associated with the formation of heterochromatin sites.
Lane 70: “HIV uses the nucleolus as a site of immune evasion since it maintains its genomic RNA; therefore, the antiviral machinery cannot detect it.” Please explain this better; it’s confusing.
Lane 80: “The nucleus is a highly dynamic structure composed of a double lipid membrane in which proteins are found that give rise to NPCs, which are responsible …” This statement needs some work; the nuclear pore complex formed by nucleoporins mediates all macromolecular transport across the nuclear envelope …
Lane 85: “The nucleus is a viscous liquid known as nucleoplasm” … the nucleus behaves as a highly viscous liquid, which affects the rate of diffusion of molecules in the nucleus.
Lane 96: “CBs are motile organelles that can pass through the nucleoplasm of cells.” … Cajal bodies are not mobile; they are physically associated with specific active gene loci that are responsible for their formation. Their restricted “movement” is primarily regulated by the loop-out genomic regions they are associated with.
Lane 118: “The nucleolus is a membrane-less organelle that can be divided into three sections” … it would be better to indicate .., three subcompartments.
Lane 121: biogenesis of pre-ribosomal subunits
Lane 124: This forms 18s, 5s, 8s, and 28s ribosomal … please correct this into … 18S, 5.8S, and 28S ribosomal RNA
Lane 130-152: The authors should reorganize these three paragraphs, focusing on the steps of rRNA synthesis, processing, and pre-ribosome assembly and combining it with tri-partite nucleolar sub-organization (see DOI:https://doi.org/10.1016/j.molcel.2019.08.014; DOI: 10.1007/s00418-024-02297-7). This will improve the flow of the text.
Lane 142: Please change “18s, 5.8s, and 28s” to 18S, 5.8S, and 28S”
Lane 151: “49 ribosomal proteins are bound with the 5S ribosomal RNA” … please, correct this statement, 49 ribosomal proteins are bound to 28S, 5.8S, and 5S rRNAs in the large 60S ribosomal subunit
Lane 159: please, remove “(ribonucleic acids)”
Lane 160: “These extra functions are because the nucleolus possesses many proteins” … this statement needs some work … many proteins are not directly involved in ribosome biogenesis but arise under various stress conditions and are associated with the nucleolar stress response (DOI: 10.1016/j.molcel.2010.09.024) …
Lane 162: “this region has three main proteins: nucleolin, fibrillarin, B23, and nucleophosmin.” … which region? The nucleolus? Please correct … B23/Nucleophosmin (this protein has two names)
TABLE 2: Please, change it to Table 1: Please indicate that SNU13, NOP56, NOP58, and Fibrillarin are the core proteins of box C/D small nucleolar ribonucleoprotein (snoRNP) particles.
Lane 178: Nucleophosmin is also involved in DNA damage repair, genomic stability, and tumorigenesis (DOI: 10.7554/eLife.13571)
Lane 182: Fibrillarin is 2'-O-methyltransferase that catalyzes RNA-base modifications on rRNA under the guidance of BOX C/D snoRNAs (https://doi.org/10.1002/cbin.11044)
Lane 194: Nucleolin is one of the most abundant nonribosomal proteins in the nucleolus
Lane 242: “during the viral cycle, it has been seen that viral proteins are generally found early in the infection in the nucleolus” … this important fact should be explained in detail
Lane 247: “viral Rev and Tat proteins can be imported into the nucleolus. Once in the nucleolus, they are retained for HIV to regulate the transcription of viral genes” … this statement must be corrected, HIV Rev protein facilitates the nuclear export of intron-containing HIV mRNAs (doi: 10.3390/v7062760)
Lane 277: “it promotes the increase of ribosomes and improves protein translation” … this should be changed to … it promotes the increase in production of ribosomes which stimulates protein synthesis
Lane 283: “viral proteins target RNA polymerase 1” … please change it to RNA polymerase I
Lane 316: please explain in detail the antiviral activity of PML nuclear bodies involved in interferon regulations
Lane 440: “Additionally, the nucleolus, a structure within the nucleus,” … please modify this statement … the nucleolus, the largest organelle in the nucleus,
Lane 458: “TABLE 1”. Please, change it to TABLE 2
Lane 459: The table should be expanded to include the nucleolar anticancer drugs listed in DOI: 10.1007/s00418-024-02297-7.
Comments on the Quality of English Language
Some edits should be provided by the editors.
Author Response
Dear Reviewers
I am pleased to resubmit for publication the revised version of “Nucleolus and its interactions with viral proteins required for successful infection”. I appreciated the constructive criticism from the associated editor and reviewers. I have addressed each of their concerns as outlined below.
Following the reviewer’s advice, I, along with my collaborators have been carefully revised and appropriate changes have been made in accordance with the reviewer’s suggestions. The responses to their comments are provided below:
We appreciate the recommendation and have contacted a service for correction of style and writing in English, through which the manuscript has been polished. In addition, a certificate of the company we hired will be attached.
Reviewer 1:
1.- Lane 36: “The canonical function of nuclear bodies is usually associated with RNA processing and ribosome biogenesis.” This statement is not entirely true and must be corrected. In principle, nuclear bodies, in general, regulate the concentration, exclusion, sequestration, assembly, modification, and recycling of specific components to regulate specific reactions involved in ribosome biogenesis, RNA transcription, and RNA processing.
We appreciate the comment, we have changed the idea to: These bodies are involved in the concentration, exclusion, sequestration, assembly, modification, and recycling of specific components involved in the regulation of ribosome biogenesis, RNA transcription, and RNA processing.
2.- Lane 66: The authors should highlight the significance of the integration of the viral genome(s) into the genome of host cells. They should discuss the generation of a double-strand break on the cellular chromosome, specific positions of the integration sites, and the virus latency associated with the formation of heterochromatin sites.
We appreciate your valuable observation and we have changes the paragraph to: During infection, some viruses integrate their genome into the host cell’s chromosomes, which can cause genetic alterations, mutagenesis, and cell death. In the case of retroviruses, such as Human Immunodeficiency virus (HIV), the integration of their genome into the host genome is a mandatory process for productive infection. Following infection, retroviruses generate linear, double-stranded cDNA through reverse transcription. This cDNA is flanked by long terminal repeats, which are part of a nucleoprotein complex known as the pre-integration complex. The pre-integration complex is composed of both viral and host cell proteins. The first step in the integration process is mediated by the retroviral integrase protein. When the pre-integration complex is located in the cytoplasm, the viral integrase hydrolyzes a dinucleotide from each 3 ́ end of the host cell’s DNA and catalyzes the asymmetric breakdown of this DNA to allow for the binding of viral DNA to this hydrolyzed end. The host cell's DNA repair machinery subsequently cleaves the 5' nucleotides that protrude from the viral DNA and fills those spaces with base pairs, stabilizing the viral genome insertion. Although all retroviruses can integrate cDNA into the host cell's DNA, the integration can differ. Gammaretroviruses and foam retroviruses, for example, are integrated into sections of DNA, where they can bind to promoter regions and activate the transcription of genes. These sites are rich in CpG islands, and are hypersensitive to DNase I activity. Conversely, lentiviruses prefer to integrate into sites where there are transcriptionally active genes. In the case of betaretroviruses, however, integration occurs randomly, and both alpharetroviruses and deltaviruses integration sites are rich in CpG islands. In the case of DNA viruses, integration has not yet been fully described in all virus types. Here, we will refer to adeno-associated virus (AAV), of which a little more is known. In AAV, the proteins responsible for the integration of viral DNA into host DNA are mediated by replication (Rep) proteins, which are endonucleases. The integration step occurs at chromosome position 19q13.42, but may also involve integration at other sites[6,7]. Viral integration can cause multiple consequences for the virus and the host cell through uncontrolled proliferation or cell death. Additionally, viral integration can activate or silence transcription, which can result in viral latency
3.- Lane 70: “HIV uses the nucleolus as a site of immune evasion since it maintains its genomic RNA; therefore, the antiviral machinery cannot detect it.” Please explain this better; it’s confusing.
We appreciate the comment, we have changed the idea and context of sentences to: Furthermore, research has shown that HIV uses the nucleolus as a site of immune evasion. Once HIV has synthesized its cDNA, it is transported into the nucleus where the immune system of the host cell is unable to detect this external genetic material.
4.- Lane 80: “The nucleus is a highly dynamic structure composed of a double lipid membrane in which proteins are found that give rise to NPCs, which are responsible …” This statement needs some work; the nuclear pore complex formed by nucleoporins mediates all macromolecular transport across the nuclear envelope …
We change the paragraph and the idea was improved: The nucleus is a dynamic structure composed of a double lipid membrane in which proteins give rise to nuclear pore complexes that connect the cytoplasm to the nucleus. Nuclear pores allow for the exchange of macromolecules between the nucleus and the cytoplasm. Therefore, they are considered important for gene regulation and cellular homeostasis as they allow selective exchange between the nucleus and the cytoplasm.
5.- Lane 85: “The nucleus is a viscous liquid known as nucleoplasm” … the nucleus behaves as a highly viscous liquid, which affects the rate of diffusion of molecules in the nucleus.
The line was changed to recommendation made reviewer
6.- Lane 96: “CBs are motile organelles that can pass through the nucleoplasm of cells.” … Cajal bodies are not mobile; they are physically associated with specific active gene loci that are responsible for their formation. Their restricted “movement” is primarily regulated by the loop-out genomic regions they are associated with.
We appreciate the comment, we have edited the idea to: These bodies are not mobile; they are physically associated with specific active gene loci that are responsible for their formation. Their restricted “movement” is primarily regulated by the loop-out genomic regions that they are associated with and are located near sites of active transcription
7.-Lane 118: “The nucleolus is a membrane-less organelle that can be divided into three sections” … it would be better to indicate .., three subcompartments.
We appreciate the observation and changes was added.
8.- Lane 121: biogenesis of pre-ribosomal subunits
We appreciate the observation and changes was added.
9.-Lane 124: This forms 18s, 5s, 8s, and 28s ribosomal … please correct this into … 18S, 5.8S, and 28S ribosomal RNA.
We appreciate the observation and changes was added.
10.-Lane 130-152: The authors should reorganize these three paragraphs, focusing on the steps of rRNA synthesis, processing, and pre-ribosome assembly and combining it with tri-partite nucleolar sub-organization (see DOI:https://doi.org/10.1016/j.molcel.2019.08.014; DOI: 10.1007/s00418-024-02297-7). This will improve the flow of the text.
We appreciate the recommendation, changes was performed and the new version is edited.
11.-Lane 142: Please change “18s, 5.8s, and 28s” to 18S, 5.8S, and 28S”
We appreciate the observation and changes was added.
12.-Lane 151: “49 ribosomal proteins are bound with the 5S ribosomal RNA” … please, correct this statement, 49 ribosomal proteins are bound to 28S, 5.8S, and 5S rRNAs in the large 60S ribosomal subunit
We appreciate the observation and changes was added.
13.-Lane 159: please, remove “(ribonucleic acids)”
We appreciate the observation and changes was added.
14.-Lane 160: “These extra functions are because the nucleolus possesses many proteins” … this statement needs some work … many proteins are not directly involved in ribosome biogenesis but arise under various stress conditions and are associated with the nucleolar stress response (DOI: 10.1016/j.molcel.2010.09.024).
We appreciate the recommendation, changes was made in the new version to: Indeed, the nucleolus is involved in additional cellular functions that may not be related to the biogenesis of ribosomal subunits, largely because more than 4500 proteins are associated with the nucleolus at different cellular stages. Additionally, the protein content of the nucleolus has been shown to be dynamic and is altered in response to cellular stress.
15,.Lane 162: “this region has three main proteins: nucleolin, fibrillarin, B23, and nucleophosmin.” … which region? The nucleolus? Please correct … B23/Nucleophosmin (this protein has two names)
We appreciate the observation and changes was added.
16.-TABLE 2: Please, change it to Table 1: Please indicate that SNU13, NOP56, NOP58, and Fibrillarin are the core proteins of box C/D small nucleolar ribonucleoprotein (snoRNP) particles.
We appreciate the observation and changes was added.
17.-Lane 178: Nucleophosmin is also involved in DNA damage repair, genomic stability, and tumorigenesis (DOI: 10.7554/eLife.13571)
We appreciate the observation and changes was added.
18.-Lane 182: Fibrillarin is 2'-O-methyltransferase that catalyzes RNA-base modifications on rRNA under the guidance of BOX C/D snoRNAs (https://doi.org/10.1002/cbin.11044)
We appreciate the observation and changes was added.
19.-Lane 194: Nucleolin is one of the most abundant nonribosomal proteins in the nucleolus
We appreciate the observation and changes was added.
20.-Lane 242: “during the viral cycle, it has been seen that viral proteins are generally found early in the infection in the nucleolus” … this important fact should be explained in detail
We appreciate your valuable observation and we have changes the paragraph to: The interaction of nucleolar and viral proteins is a process that allows viruses to replicate, transcribe, and, in some cases, assemble and transport their viral particles. When viral proteins enter the nucleolus, they compete with the resident proteins of the nucleolus, which generates an imbalance of the existing structures and the functions that these structures carry out in a "normal" way. This leads to the alteration of processes, such as ribosome biogenesis, cell cycle, and cell death.
21.-Lane 247: “viral Rev and Tat proteins can be imported into the nucleolus. Once in the nucleolus, they are retained for HIV to regulate the transcription of viral genes” … this statement must be corrected, HIV Rev protein facilitates the nuclear export of intron-containing HIV mRNAs (doi: 10.3390/v7062760)
We appreciate the observation and changes was added.
22.-Lane 277: “it promotes the increase of ribosomes and improves protein translation” … this should be changed to … it promotes the increase in production of ribosomes which stimulates protein synthesis
We appreciate the observation and changes was added.
23.- Lane 283: “viral proteins target RNA polymerase 1” … please change it to RNA polymerase I
We appreciate the observation and changes was added.
24.-Lane 316: please explain in detail the antiviral activity of PML nuclear bodies involved in interferon regulations
We appreciate the observation and changes was added in the figure legend.
25.- Lane 440: “Additionally, the nucleolus, a structure within the nucleus,” … please modify this statement … the nucleolus, the largest organelle in the nucleus, Lane 458: “TABLE 1”. Please, change it to TABLE 2 Lane 459: The table should be expanded to include the nucleolar anticancer drugs listed in DOI: 10.1007/s00418-024-02297-7.
We appreciate the observation and changes was added.
Reviewer 2
1.- The paper provides an overview of nuclear bodies, traditionally including the nucleolus, Cajal bodies (CBs), and promyelocytic leukemia nuclear bodies. Authors should expand their discussion to include additional nuclear bodies such as paraspeckles, speckles, super-enhancer condensates, or initiation Pol2 condensates. For a complete understanding, reference to comprehensive reviews is recommended (PMID: 28577509). In addition, a better introduction to the role of viruses would benefit from mentioning the review (PMID: 36905213). The paper neglects the role of nuclear phosphoinositides in the formation of nuclear bodies and their importance. Including references to papers (PMID: 37973889, PMID: 38734927, PMID: 30344928, PMID: 30249540, PMID: 36979361) would enhance the discussion of this topic. The figures in the manuscript need improvement for clarity. In particular, Figure 2 needs an enlarged caption and labeled sections (A, B, C, etc.) to facilitate understanding. This recommendation applies to other figures in the paper to improve their readability and interpretability. The description of drug effects is insufficient. For instance, doxorubicin is known to induce double-strand DNA breaks, and 5-fluorouracil (5FU) incorporates into RNA during transcription. The authors should provide a detailed explanation of these mechanisms and include additional examples. The effects of leptomycin and other drugs mentioned should be thoroughly discussed, highlighting their molecular targets and mechanisms of action. The manuscript suffers from poor English and requires extensive editing for grammatical correctness and coherence. The overall quality of the writing needs significant improvement to ensure clarity.
We appreciate your comments and recommendations. We would like to inform you that we have modified the new version by rethinking the ideas and including new bibliography. You will be able to identify these changes in the corrected version, highlighted in yellow.
Finally, we again thank you for your suggestions and insights, which have enriched the manuscript and produced a more balanced and better account of the review. We hope that the revised manuscript is now suitable for publication in the prestigious journal that you represent.
I look forward to your reply.
Sincerely,
Dr. Moises Leon Juarez
Dr. Luis Adrian de Jesus Gonzalez

Reviewer 2 Report
Comments and Suggestions for Authors
The paper provides an overview of nuclear bodies, traditionally including the nucleolus, Cajal bodies (CBs), and promyelocytic leukemia nuclear bodies. Authors should expand their discussion to include additional nuclear bodies such as paraspeckles, speckles, super-enhancer condensates, or initiation Pol2 condensates. For a complete understanding, reference to comprehensive reviews is recommended (PMID: 28577509). In addition, a better introduction to the role of viruses would benefit from mentioning the review (PMID: 36905213). The paper neglects the role of nuclear phosphoinositides in the formation of nuclear bodies and their importance. Including references to papers (PMID: 37973889, PMID: 38734927, PMID: 30344928, PMID: 30249540, PMID: 36979361) would enhance the discussion of this topic. The figures in the manuscript need improvement for clarity. In particular, Figure 2 needs an enlarged caption and labeled sections (A, B, C, etc.) to facilitate understanding. This recommendation applies to other figures in the paper to improve their readability and interpretability. The description of drug effects is insufficient. For instance, doxorubicin is known to induce double-strand DNA breaks, and 5-fluorouracil (5FU) incorporates into RNA during transcription. The authors should provide a detailed explanation of these mechanisms and include additional examples. The effects of leptomycin and other drugs mentioned should be thoroughly discussed, highlighting their molecular targets and mechanisms of action. The manuscript suffers from poor English and requires extensive editing for grammatical correctness and coherence. The overall quality of the writing needs significant improvement to ensure clarity.
In summary, while the paper touches on several important aspects of nuclear bodies and viral interactions, it requires substantial revision for accuracy, clarity, and completeness. Enhancing the discussion with recent literature to provide context, improving figure captions, and providing detailed descriptions of drug effects will significantly strengthen the manuscript. In addition, thorough language editing is essential to improve the quality of the writing.
Comments on the Quality of English LanguageThe manuscript contains numerous problems with the English language, including grammatical errors and awkward sentence structures. These problems detract from the clarity and readability of the text, making it difficult for readers to fully understand the content and thus to evaluate the manuscript responsibly. To improve the quality of the manuscript, a thorough and comprehensive editing process is required. This process should focus on improving sentence flow, coherence, and overall clarity. Addressing this issue is essential to raising the quality of writing to a level that is clear and easily understood.
Author Response
Dear Reviewers
I am pleased to resubmited for publication the revised version of “Nucleolus and its interactions with viral proteins required for successful infection”. I appreciated the constructive criticism from the associated editor and reviewers. I have addressed each of their concerns as outlined below.
Following the reviewer’s advice, I, along with my collaborators have been carefully revised and appropriate changes have been made in accordance with the reviewer’s suggestions. The responses to their comments are provided below:
We appreciate the recommendation and have contacted a service for correction of style and writing in English, through which the manuscript has been polished. In addition, a certificate of the company we hired will be attached.
Reviewer 1:
1.- Lane 36: “The canonical function of nuclear bodies is usually associated with RNA processing and ribosome biogenesis.” This statement is not entirely true and must be corrected. In principle, nuclear bodies, in general, regulate the concentration, exclusion, sequestration, assembly, modification, and recycling of specific components to regulate specific reactions involved in ribosome biogenesis, RNA transcription, and RNA processing.
We appreciate the comment, we have changed the idea to: These bodies are involved in the concentration, exclusion, sequestration, assembly, modification, and recycling of specific components involved in the regulation of ribosome biogenesis, RNA transcription, and RNA processing.
2.- Lane 66: The authors should highlight the significance of the integration of the viral genome(s) into the genome of host cells. They should discuss the generation of a double-strand break on the cellular chromosome, specific positions of the integration sites, and the virus latency associated with the formation of heterochromatin sites.
We appreciate your valuable observation and we have changes the paragraph to: During infection, some viruses integrate their genome into the host cell’s chromosomes, which can cause genetic alterations, mutagenesis, and cell death. In the case of retroviruses, such as Human Immunodeficiency virus (HIV), the integration of their genome into the host genome is a mandatory process for productive infection. Following infection, retroviruses generate linear, double-stranded cDNA through reverse transcription. This cDNA is flanked by long terminal repeats, which are part of a nucleoprotein complex known as the pre-integration complex. The pre-integration complex is composed of both viral and host cell proteins. The first step in the integration process is mediated by the retroviral integrase protein. When the pre-integration complex is located in the cytoplasm, the viral integrase hydrolyzes a dinucleotide from each 3 ́ end of the host cell’s DNA and catalyzes the asymmetric breakdown of this DNA to allow for the binding of viral DNA to this hydrolyzed end. The host cell's DNA repair machinery subsequently cleaves the 5' nucleotides that protrude from the viral DNA and fills those spaces with base pairs, stabilizing the viral genome insertion. Although all retroviruses can integrate cDNA into the host cell's DNA, the integration can differ. Gammaretroviruses and foam retroviruses, for example, are integrated into sections of DNA, where they can bind to promoter regions and activate the transcription of genes. These sites are rich in CpG islands, and are hypersensitive to DNase I activity. Conversely, lentiviruses prefer to integrate into sites where there are transcriptionally active genes. In the case of betaretroviruses, however, integration occurs randomly, and both alpharetroviruses and deltaviruses integration sites are rich in CpG islands. In the case of DNA viruses, integration has not yet been fully described in all virus types. Here, we will refer to adeno-associated virus (AAV), of which a little more is known. In AAV, the proteins responsible for the integration of viral DNA into host DNA are mediated by replication (Rep) proteins, which are endonucleases. The integration step occurs at chromosome position 19q13.42, but may also involve integration at other sites[6,7]. Viral integration can cause multiple consequences for the virus and the host cell through uncontrolled proliferation or cell death. Additionally, viral integration can activate or silence transcription, which can result in viral latency
3.- Lane 70: “HIV uses the nucleolus as a site of immune evasion since it maintains its genomic RNA; therefore, the antiviral machinery cannot detect it.” Please explain this better; it’s confusing.
We appreciate the comment, we have changed the idea and context of sentences to: Furthermore, research has shown that HIV uses the nucleolus as a site of immune evasion. Once HIV has synthesized its cDNA, it is transported into the nucleus where the immune system of the host cell is unable to detect this external genetic material.
4.- Lane 80: “The nucleus is a highly dynamic structure composed of a double lipid membrane in which proteins are found that give rise to NPCs, which are responsible …” This statement needs some work; the nuclear pore complex formed by nucleoporins mediates all macromolecular transport across the nuclear envelope …
We change the paragraph and the idea was improved: The nucleus is a dynamic structure composed of a double lipid membrane in which proteins give rise to nuclear pore complexes that connect the cytoplasm to the nucleus. Nuclear pores allow for the exchange of macromolecules between the nucleus and the cytoplasm. Therefore, they are considered important for gene regulation and cellular homeostasis as they allow selective exchange between the nucleus and the cytoplasm.
5.- Lane 85: “The nucleus is a viscous liquid known as nucleoplasm” … the nucleus behaves as a highly viscous liquid, which affects the rate of diffusion of molecules in the nucleus.
The line was changed to recommendation made reviewer
6.- Lane 96: “CBs are motile organelles that can pass through the nucleoplasm of cells.” … Cajal bodies are not mobile; they are physically associated with specific active gene loci that are responsible for their formation. Their restricted “movement” is primarily regulated by the loop-out genomic regions they are associated with.
We appreciate the comment, we have edited the idea to: These bodies are not mobile; they are physically associated with specific active gene loci that are responsible for their formation. Their restricted “movement” is primarily regulated by the loop-out genomic regions that they are associated with and are located near sites of active transcription
7.-Lane 118: “The nucleolus is a membrane-less organelle that can be divided into three sections” … it would be better to indicate .., three subcompartments.
We appreciate the observation and changes was added.
8.- Lane 121: biogenesis of pre-ribosomal subunits
We appreciate the observation and changes was added.
9.-Lane 124: This forms 18s, 5s, 8s, and 28s ribosomal … please correct this into … 18S, 5.8S, and 28S ribosomal RNA.
We appreciate the observation and changes was added.
10.-Lane 130-152: The authors should reorganize these three paragraphs, focusing on the steps of rRNA synthesis, processing, and pre-ribosome assembly and combining it with tri-partite nucleolar sub-organization (see DOI:https://doi.org/10.1016/j.molcel.2019.08.014; DOI: 10.1007/s00418-024-02297-7). This will improve the flow of the text.
We appreciate the recommendation, changes was performed and the new version is edited.
11.-Lane 142: Please change “18s, 5.8s, and 28s” to 18S, 5.8S, and 28S”
We appreciate the observation and changes was added.
12.-Lane 151: “49 ribosomal proteins are bound with the 5S ribosomal RNA” … please, correct this statement, 49 ribosomal proteins are bound to 28S, 5.8S, and 5S rRNAs in the large 60S ribosomal subunit
We appreciate the observation and changes was added.
13.-Lane 159: please, remove “(ribonucleic acids)”
We appreciate the observation and changes was added.
14.-Lane 160: “These extra functions are because the nucleolus possesses many proteins” … this statement needs some work … many proteins are not directly involved in ribosome biogenesis but arise under various stress conditions and are associated with the nucleolar stress response (DOI: 10.1016/j.molcel.2010.09.024).
We appreciate the recommendation, changes was made in the new version to: Indeed, the nucleolus is involved in additional cellular functions that may not be related to the biogenesis of ribosomal subunits, largely because more than 4500 proteins are associated with the nucleolus at different cellular stages. Additionally, the protein content of the nucleolus has been shown to be dynamic and is altered in response to cellular stress.
15,.Lane 162: “this region has three main proteins: nucleolin, fibrillarin, B23, and nucleophosmin.” … which region? The nucleolus? Please correct … B23/Nucleophosmin (this protein has two names)
We appreciate the observation and changes was added.
16.-TABLE 2: Please, change it to Table 1: Please indicate that SNU13, NOP56, NOP58, and Fibrillarin are the core proteins of box C/D small nucleolar ribonucleoprotein (snoRNP) particles.
We appreciate the observation and changes was added.
17.-Lane 178: Nucleophosmin is also involved in DNA damage repair, genomic stability, and tumorigenesis (DOI: 10.7554/eLife.13571)
We appreciate the observation and changes was added.
18.-Lane 182: Fibrillarin is 2'-O-methyltransferase that catalyzes RNA-base modifications on rRNA under the guidance of BOX C/D snoRNAs (https://doi.org/10.1002/cbin.11044)
We appreciate the observation and changes was added.
19.-Lane 194: Nucleolin is one of the most abundant nonribosomal proteins in the nucleolus
We appreciate the observation and changes was added.
20.-Lane 242: “during the viral cycle, it has been seen that viral proteins are generally found early in the infection in the nucleolus” … this important fact should be explained in detail
We appreciate your valuable observation and we have changes the paragraph to: The interaction of nucleolar and viral proteins is a process that allows viruses to replicate, transcribe, and, in some cases, assemble and transport their viral particles. When viral proteins enter the nucleolus, they compete with the resident proteins of the nucleolus, which generates an imbalance of the existing structures and the functions that these structures carry out in a "normal" way. This leads to the alteration of processes, such as ribosome biogenesis, cell cycle, and cell death.
21.-Lane 247: “viral Rev and Tat proteins can be imported into the nucleolus. Once in the nucleolus, they are retained for HIV to regulate the transcription of viral genes” … this statement must be corrected, HIV Rev protein facilitates the nuclear export of intron-containing HIV mRNAs (doi: 10.3390/v7062760)
We appreciate the observation and changes was added.
22.-Lane 277: “it promotes the increase of ribosomes and improves protein translation” … this should be changed to … it promotes the increase in production of ribosomes which stimulates protein synthesis
We appreciate the observation and changes was added.
23.- Lane 283: “viral proteins target RNA polymerase 1” … please change it to RNA polymerase I
We appreciate the observation and changes was added.
24.-Lane 316: please explain in detail the antiviral activity of PML nuclear bodies involved in interferon regulations
We appreciate the observation and changes was added in the figure legend.
25.- Lane 440: “Additionally, the nucleolus, a structure within the nucleus,” … please modify this statement … the nucleolus, the largest organelle in the nucleus, Lane 458: “TABLE 1”. Please, change it to TABLE 2 Lane 459: The table should be expanded to include the nucleolar anticancer drugs listed in DOI: 10.1007/s00418-024-02297-7.
We appreciate the observation and changes was added.
Reviewer 2
1.- The paper provides an overview of nuclear bodies, traditionally including the nucleolus, Cajal bodies (CBs), and promyelocytic leukemia nuclear bodies. Authors should expand their discussion to include additional nuclear bodies such as paraspeckles, speckles, super-enhancer condensates, or initiation Pol2 condensates. For a complete understanding, reference to comprehensive reviews is recommended (PMID: 28577509). In addition, a better introduction to the role of viruses would benefit from mentioning the review (PMID: 36905213). The paper neglects the role of nuclear phosphoinositides in the formation of nuclear bodies and their importance. Including references to papers (PMID: 37973889, PMID: 38734927, PMID: 30344928, PMID: 30249540, PMID: 36979361) would enhance the discussion of this topic. The figures in the manuscript need improvement for clarity. In particular, Figure 2 needs an enlarged caption and labeled sections (A, B, C, etc.) to facilitate understanding. This recommendation applies to other figures in the paper to improve their readability and interpretability. The description of drug effects is insufficient. For instance, doxorubicin is known to induce double-strand DNA breaks, and 5-fluorouracil (5FU) incorporates into RNA during transcription. The authors should provide a detailed explanation of these mechanisms and include additional examples. The effects of leptomycin and other drugs mentioned should be thoroughly discussed, highlighting their molecular targets and mechanisms of action. The manuscript suffers from poor English and requires extensive editing for grammatical correctness and coherence. The overall quality of the writing needs significant improvement to ensure clarity.
We appreciate your comments and recommendations. We would like to inform you that we have modified the new version by rethinking the ideas and including new bibliography. You will be able to identify these changes in the corrected version, highlighted in yellow.
Finally, we again thank you for your suggestions and insights, which have enriched the manuscript and produced a more balanced and better account of the review. We hope that the revised manuscript is now suitable for publication in the prestigious journal that you represent.
I look forward to your reply.
Sincerely,
Dr. Moises Leon Juarez
Dr. Luis Adrian De Jesus Gonzalez

Round 2
Reviewer 1 Report
Comments and Suggestions for Authors
The revised manuscript is suitable for the publication.
Reviewer 2 Report
Comments and Suggestions for Authors
The authors have extensively rewritten the manuscript and corrected the major issues raised by the reviewer. The readability of the English has been improved. The figures now have appropriately sized captions and descriptions. After resubmission of the corrected version, I believe that it is ready for publication in Cells Journal.